# Quantitative Rapid Test for Detection and Monitoring of Active Pulmonary Tuberculosis in Nonhuman Primates

**DOI:** 10.3390/biology10121260

**Published:** 2021-12-02

**Authors:** Zijie Zhou, Anouk van Hooij, Richard Vervenne, Claudia C. Sombroek, Elisa M. Tjon Kon Fat, Tom H. M. Ottenhoff, Paul L. A. M. Corstjens, Frank Verreck, Annemieke Geluk

**Affiliations:** 1Department of Infectious Diseases, Leiden University Medical Center, 2333 ZA Leiden, The Netherlands; z.zhou@lumc.nl (Z.Z.); a.van_hooij@lumc.nl (A.v.H.); t.h.m.ottenhoff@lumc.nl (T.H.M.O.); 2Section of TB Research & Immunology, Biomedical Primate Research Center (BPRC), 2288 GJ Rijswijk, The Netherlands; vervenne@bprc.nl (R.V.); sombroek@bprc.nl (C.C.S.); verreck@bprc.nl (F.V.); 3Department of Cell and Chemical Biology, Leiden University Medical Center, 2333 ZA Leiden, The Netherlands; e.m.tjon_kon_fat@lumc.nl (E.M.T.K.F.); p.l.a.m.corstjens@lumc.nl (P.L.A.M.C.)

**Keywords:** biomarkers, diagnosis, nonhuman primates (NHPs), lateral flow assay, up-converting reporter particles, MTB, tuberculosis

## Abstract

**Simple Summary:**

Tuberculosis (TB), caused by *Mycobacterium tuberculosis* (MTB), is the most lethal infectious disease from a single pathogen for which there is no effective vaccine available. Rhesus macaques are extremely susceptible to MTB and therefore represent relevant models to study the pathogenesis of TB and assess the potential of TB drugs and vaccines. However, there are no diagnostic tools currently available that allow rapid, user-friendly detection of TB for either TB research purposes or monitoring nonhuman primate colonies. To develop a rapid diagnostic test, we investigated whether low complexity lateral flow assays (LFAs) that we recently developed for rapid and quantitative detection of human serum proteins are applicable to detect and monitor active pulmonary TB in NHPs. We found that serum levels of SAA1, IP-10, and IL-6 detected by LFAs were significantly increased after MTB infection in rhesus macaques. Moreover, levels of these biomarkers correlated with disease severity as determined by pathology scoring and allowed detection of the effect of vaccination and drug treatment in experimentally MTB infected macaques. These UCP-LFAs thus offer a low-cost, convenient, and minimally invasive diagnostic tool that can be used to assess new therapeutic and prophylactic treatment methods in macaques to tackle TB.

**Abstract:**

Nonhuman primates (NHPs) are relevant models to study the pathogenesis of tuberculosis (TB) and evaluate the potential of TB therapies, but rapid tools allowing diagnosis of active pulmonary TB in NHPs are lacking. This study investigates whether low complexity lateral flow assays utilizing upconverting reporter particles (UCP-LFAs) developed for rapid detection of human serum proteins can be applied to detect and monitor active pulmonary TB in NHPs. UCP-LFAs were used to assess serum proteins levels and changes in relation to the MTB challenge dosage, lung pathology, treatment, and disease outcome in experimentally MTB-infected macaques. Serum levels of SAA1, IP-10, and IL-6 showed a significant increase after MTB infection in rhesus macaques and correlated with disease severity as determined by pathology scoring. Moreover, these biomarkers could sensitively detect the reduction of bacterial levels in the lungs of macaques due to BCG vaccination or drug treatment. Quantitative measurements by rapid UCP-LFAs specific for SAA1, IP-10, and IL-6 in serum can be utilized to detect active progressive pulmonary TB in macaques. The UCP-LFAs thus offer a low-cost, convenient, and minimally invasive diagnostic tool that can be applied in studies on TB vaccine and drug development involving macaques.

## 1. Introduction

Tuberculosis (TB) remains one of the deadliest infectious diseases from a single pathogen worldwide. In 2019 the World Health Organization (WHO) reported 10 million people who developed TB, and an estimated 1.4 million TB-related deaths among human immunodeficiency virus (HIV)-negative people, and an additional 208,000 among HIV-positive people [1].

Among the animal models for TB research, nonhuman primates (NHPs), particularly rhesus macaques (*Macaca mulatta*) and cynomolgus macaques (*Macaca fascicularis*) bear a remarkable resemblance to humans with respect to genetics, anatomy, physiology, and immunology [2,3,4,5,6]. Upon *M. tuberculosis* (MTB) infection, rhesus macaques appear highly susceptible, while cynomolgus macaques are more resistant to disease development [7], showing similar clinical and pathological manifestations as in human TB [8,9,10,11]. These features contribute to the validity of macaques as suitable animal models for studying pathological characteristics of TB, dynamics of disease progression, and the efficacy of new therapies and vaccines for TB.

Within the ethical restrictions of using NHPs for research, specific colonies provide purpose-bred animals of the second (F2) or higher filial generations. Concerning their susceptibility for TB [7,11,12], screening of such NHP breeding colonies in research centers as well as in zoos or wildlife parks is essential for monitoring possible anthroponosis or zoonosis and preventing TB outbreaks through intra- and interspecies transmission [13,14,15]. The most commonly used diagnostic method for TB in humans [16]—microscopic examination of sputum smears—detects advanced stages of infection only and is impractical in surveillance of NHPs while collecting saliva (sputum) from NHPs is not trivial. Currently, the typical screening method for monitoring TB in NHPs is the palpebral tuberculin skin test (TST), relying on the readout by visual inspection of a type IV hypersensitivity response that typically occurs if there has been a prior infection with MTB [17,18]. As in humans, TST screening in NHPs is compromised by several aspects, such as reader variability of the hypersensitivity response, the occurrence of false positive results due to cross-reactivity of the TST with other nonpathogenic mycobacteria [19,20,21], and relatively low sensitivity with a risk of false negative results [3,22,23,24]. The use of in vitro IFN-γ release assays (IGRAs), such as Primagam^®^ [22], requires venepuncture, typically under sedation, and specific lab equipment that, apart from dedicated research centers, is often not available in zoos or wildlife parks. Similar restrictions apply to molecular detection of MTB (e.g., by Gene Xpert) and conventional X-ray diagnostics, especially to more advanced imaging systems such as PET-CT. Therefore, more practical modalities for the diagnosis of TB applicable to laboratory resource-limited environments would help relieve and provide relevant veterinary care and colony management support. 

Recent progress in rapid diagnostics development for human mycobacterial disease [25,26,27,28,29] urges us to investigate the potential value of quantitative host biomarker detection in NHPs from a low invasive, small-sized biosample, such as fingerstick blood. The application of up-converting reporter particle (UCP) technology in the robust and low complexity, user-friendly lateral flow (LF) format has been previously shown to provide a reliable method for quantitative detection of human cytokines and antibodies in serum as a point-of-care (POC) alternative for laboratory-based enzyme-linked immunosorbent assay (ELISA) [25,26,27,30,31,32,33,34]. 

As has become clear from biomarker research for mycobacterial diseases TB and leprosy, the use of combined analysis of biomarkers through a biomarker signature improves sensitivity and specificity for diagnosis [28,35,36]. In this respect, host protein biomarker signatures were identified that distinguished active, pulmonary TB patients from other respiratory diseases (ORD) amongst primary healthcare clinic attendees in Africa with signs and symptoms suggestive of TB [37,38,39]. This prompted us to apply the most promising, corresponding host protein markers from this signature: apolipoprotein A1 (ApoA1), C reactive protein (CRP), ferritin, IFN-γ-inducible protein 10 (IP-10), and serum amyloid A (SAA), as well as the general infection marker interleukin 6 (IL-6) to our UCP-LFA format.

Since reagents against human proteins have been reported to cross-react with NHP orthologues [5,6,8,26,27,33,34,40,41], the UCP-LFAs developed for monitoring TB in humans are also potentially suitable to monitor TB in NHPs.

In this exploratory study, we aimed to determine whether previously developed UCP-LFAs for the detection of various biomarkers for human pulmonary TB are applicable to detect TB as well as monitor disease progression in NHPs. UCP-LFAs for 11 human proteins were used to quantitatively determine host protein levels in biobanked sera from experimentally MTB-infected, and BCG vaccinated macaques. To assess the diagnostic potential of the UCP-LFA in NHPs, measured biomarker levels were related to disease status as well as pathology of the macaques.

## 2. Materials and Methods

### 2.1. Ethics

The present studies were performed on NHP serum samples banked from previous MTB infection experiments at BPRC. For these studies, ethical clearance was obtained from the independent ethical authority prior to the start, following Dutch law. The respective studies from which samples were used were registered under dossier numbers DEC579, DEC671, DEC690, CCD009A, CCD009B, and DEC761subA. All housing and animal care procedures complied with European directive 2010/63/EU, as well as the “Standard for Humane Care and Use of Laboratory Animals by Foreign Institutions” provided by the Department of Health and Human Services of the US National Institutes of Health (NIH, identification number A5539-01). Since 2012, BPRC is AAALAC accredited.

All NHP TB studies were limited in time and had fixed endpoints by the ethically approved protocols. Humane endpoint criteria were predefined to avoid unwanted levels of discomfort. At the endpoint being premature (by reaching humane endpoint criteria or defined by the experimental protocol), all animals were sedated, euthanized, and submitted to pathological postmortem evaluation. 

### 2.2. NHP Study Cohorts

The NHP cohorts from which serum samples were analyzed (Appendix A) comprised rhesus macaques (*Macaca mulatta*) of Chinese or Indian genotype and cynomolgus macaques (*Macaca fascicularis*) [42]. Animals were experimentally infected with MTB strain Erdman K01 (obtained through BEI Resources) by endobronchial instillation. Macaques were challenged either with a high dose of 500 colony forming units (CFU) of MTB, a low dose of 15 CFU, or an ultralow dose of 1 to 7 CFU. As indicated, rhesus macaques in the high-dose MTB challenge cohort had or had not been vaccinated with BCG Danish 1331 (Statens Serum Institute, Denmark) 38 or 17 weeks before infection. A group of six rhesus macaques were infected with 15 CFU/dose MTB and subsequently treated with 30 mg/kg rifampin (RIF) and 15 mg/kg isoniazid (INH) (RIF + INH) daily from week 4 to week 12 after the infection. For all study groups, serum samples were collected at the indicated time points before and after the MTB challenge.

TB pathology was quantified postmortem using an arbitrary, semiquantitative grading system for the size, manifestation, and frequency of lesions and for lymph node involvement as previously published [43,44]. Lung lobes were cut in slices of approximately 5 mm prior to pathology scoring.

### 2.3. UCP Conjugates

LFAs were performed with luminescent up-converting reporter particles (UCP) as a sensitive background-free label for quantitative detection of targeted analytes [45,46,47]. UCP nanomaterials (200 nm NaYF4:Yb^3+^, Er^3+^ particles, functionalized with carboxyl groups) were obtained from Intelligent Material Solutions Inc. (Princeton, NJ, USA). For chemokine (C-C motif) ligand 4 (CCL4), interleukin-1 receptor antagonist (IL-1Ra), tumor necrosis factor (TNF), S100 calcium-binding protein A12 (S100A12), ApoA1, IP-10, and CRP, the UCP-conjugates and complimentary LF strips were produced as described previously [27,48]. For serum amyloid A1 (SAA1), interleukin 6 (IL-6), ferritin, and complement component 1q (C1q), UCP conjugates were prepared with mouse antihuman Serum Amyloid A1 (SAA1, Novus Biologicals, Centennial, CO, USA), mouse antihuman Ferritin mAb (F23; Novus biologicals, Littleton, CO, USA), or mouse antihuman C1q (2204, LUMC, Leiden, The Netherlands) at a concentration of 50 μg antibody per mg UCP, and the rat antihuman-IL-6 (MQ2-13A5, BioLegend, San Diego, CA, USA) at a concentration of 25 μg antibody per mg UCP according to the method described previously [25]. 

### 2.4. Lateral Flow (LF) Strips

LF strips were made as described previously [27,31,34,48]. For SAA1, IL-6, Ferritin, and C1q, the test (T) line comprised mouse-anti-human SAA1 (SAA1, Novus Biologicals, Centennial, CO, USA), rat antihuman-IL-6 (MQ2-39C3, BioLegend, San Diego, CA, USA), mouse antihuman Ferritin mAb (F31; Novus biologicals, Littleton, USA), or mouse antihuman C1q (2204, LUMC, Leiden, The Netherlands), respectively, at a concentration of 200 ng per 4 mm width. The flow-control (FC line) of ferritin, C1q, and SAA1 comprised goat-anti-mouse IgG antibody (M8642; Sigma-Aldrich, St. Louis, MO, USA), and the FC line of IL-6 comprised goat-anti-rat IgG antibody (R5130; Sigma-Aldrich). UCP reporter conjugate was applied to the sample/conjugate-release pad at a density of 400 ng per 4 mm width in a buffer containing 5% (*w*/*v*) sucrose, 50 mM Tris pH 8.0, 0.6 mM Borate pH 8, 135 mM NaCl, 0.5% (*w*/*v*) BSA, and 0.25% Tween-20. The pads were dried for 1 h at 37 °C.

### 2.5. UCP-LFAs

Serum samples were diluted in high salt finger stick (HSFS) buffer (100 mM Tris pH 8, 270 mM NaCl, 1% (*w*/*v*) BSA, 1% (*v*/*v*) Triton X-100) as follows: 1:5 (for CCL4, IL-1Ra, TNF, S100A12, SAA1, IP-10, IL-6, ferritin), 1:50 (for CRP, C1q), or 1:500 (for ApoA1). The diluted serum samples (50 μL) were applied to the LF strips, which were analyzed (after drying) using a UCP dedicated benchtop reader (UPCON; Labrox, Finland). Results are displayed as the Ratio (R) of T over FC signals (peak area) based on relative fluorescence units (RFUs).

### 2.6. Statistical Analysis

Graphpad Prism version 9.0 for Windows (GraphPad Software, San Diego, CA, USA) was used to perform Mann-Whitney U tests, Wilcoxon matched-pairs signed rank tests, Kruskal-Wallis with Dunn’s correction for multiple testing, plot receiver operating characteristic (ROC) curves, calculate the area under the curve (AUC), plot Kaplan–Meier curves, compare the curves by Mantel-Cox test, and compute Spearman correlation coefficients. The optimal sensitivity and specificity were determined using the Youden’s index [49]. The statistical significance level used was *p* < 0.05.

## 3. Results

### 3.1. Detection of MTB Infection in Rhesus Macaques Using UCP-LFAs for Human Serum Proteins

To investigate whether UCP-LFAs previously developed for human serum proteins allow detection of homologous proteins in sera of NHPs, we first piloted a panel of 11 host proteins using a limited number of samples from rhesus macaques that had been infected with a high dose of MTB. Levels of SAA1, IL-6, and IP-10 were found to increase after infection (Appendix A). All other markers (CCL4, IL-1Ra, TNF, S100A12, ferritin, C1q, CRP, and ApoA1) yielded no or hardly any detectable signals (Appendix A) or failed to discriminate before and after infection (ApoA1, C1q) (Appendix A). Thus, we continued our efforts with the three former, most discriminative UCP-LFAs.

To assess the diagnostic potential of UCP-LFAs for human SAA1, IP-10, and IL-6 for MTB infection in NHP, we analyzed banked serum samples from cohorts of previous TB research projects (Appendix A). All animals had been confirmed to have manifest granulomatous pulmonary TB by pathological and bacteriological postmortem evaluation (Appendix A). As the diagnostic readout for 75 rhesus macaques, the ratio values (R) of relative fluorescence units (RFU) from test over flow-control (T/FC) areas of the UCP-LFA were determined (Figure 1a): postinfection signals were significantly elevated for SAA1, IP-10, as well as IL-6 (Figure 1a). ROC-AUC values of 0.82, 0.84, and 0.79 were obtained for SAA1, IP-10, and IL-6, respectively, demonstrating that all three UCP-LFAs discriminated pre from postinfection NHP serum samples well (Figure 1b, Table 1a). Notably, when the three biomarkers (3BM) were combined, the diagnostic performance was optimal (ROC-AUC: 0.87; sensitivity: 78.67%; specificity: 89.33%).

While samples had been obtained from TB vaccine research studies in NHP, reflecting treatment conditions that are not applied in regular breeding or wildlife cohorts, we also performed a meta-analysis on the UCP-LFA data from untreated (infection) controls only. Furthermore, we excluded samples obtained from animals that had received an ultra-low infectious challenge dose of one to seven colony-forming units of MTB or a short follow-up time of six weeks postinfection only (Appendix A). The ratios and the ROC curves are plotted in Figure 1c,d, respectively, and the UCP-LFA performance characteristics are summarized in Table 1b. As expected (based on protective effects of prior vaccination suppressing disease manifestation), sensitivity increased when only serum samples from unvaccinated control animals after a single dose infection and postinfection follow-up of at least two months were considered, generating ROC-AUC values of 0.87, 0.95, 0.82 and 0.91 for SAA1, IP-10, IL-6, and 3BM, respectively (Table 1b). It is noted that IP-10 showed the highest sensitivity, while the specificity of SAA1 and IL-6 were higher than that obtained with IP-10 (Table 1). These data, thus, underpin the TB diagnostic potential of our UCP-LFA platforms in NHP.

### 3.2. Diagnostic Performance Relative to Infection Dose

It is imperative that experimental infection dose impacts disease kinetics and the ultimate level of pathological involvement at the endpoint. Therefore, we analyzed results by comparing MTB challenge doses of 500 CFU/dose versus 15 CFU/dose separately. First, we focused on the diagnostic performance by considering the endpoint samples of these untreated infection control animals. At the endpoint, the lung pathology scores had been found higher in the high-dose than in the low-dose infections (Appendix A). The UCP-LFA performance of the high- vs. low-dose MTB challenge is illustrated in a heatmap in Appendix A. The proportion of SAA1 positive samples appeared 83.3% in the high challenge dose versus 58.3% in the low-dose challenge group. The reduction in diagnostic capacity from high- to low-dose infection was less dramatic with IP-10 (91.7% to 75% positivity). For IL-6, however, it was even more pronounced than with SAA1, as it went from 83.3% in high-dose to 25% in low-dose infected rhesus macaques (*p* < 0.01) (Appendix A). Furthermore, the combination of the three biomarkers (3BM; Appendix A) resulted in more positive tests in the high-dose than low-dose infection group (median: 3 vs. 2, *p* < 0.01).

### 3.3. Diagnostic Performance Relative to Time after Infection

As a next step, we interrogated the kinetics of UCP-LFA diagnostics by analyzing longitudinal serum samples that had been banked at two to three weekly intervals from the infectious challenge time point up to 12 weeks postinfection. Again, as in the previous paragraph, we focused on untreated infection controls only and analyzed the high- and low-dose MTB challenge groups separately. The levels of SAA1 showed an immediate increase within 3–4 weeks after MTB infection, independent of challenge dose and leveling off at subsequent time points (Figure 2a). The IP-10, on the contrary, gradually increased until the study endpoint, while IL-6, in general, gave weaker signals with a less pronounced kinetic profile (Figure 2b,c). Of note, the three biomarkers showed significant but not particularly high correlations with each other (Appendix A). This is clear also from the finding that these biomarkers showed the highest levels in high-dose infected rhesus macaques (with a significant difference from the low-dose group) at week six for SAA1, at week 12 for IP-10, and at week 6 and 12 for IL-6. The performance of the respective markers and 3BM, all expressed as ROC-AUC values over time, is listed for high- and low-dose infected rhesus macaques in Table 2a,b, respectively. Although the superior sensitivity of IP-10 is corroborated by this ROC-AUC time analysis and in the low dose infection group in particular, it is remarkable that in the high-dose challenge cohort, 3BM diagnostics provided the highest ROC-AUC values at three out of four time points (Table 2).

### 3.4. Correlation between UCP-LFA Signals and TB Pathology

To formalise that SAA1, IP-10 and IL-6 levels reflect disease severity we performed a statistical correlation analysis between diagnostic and pathological endpoint measures. Pathological involvement was determined in arbitrary units upon postmortem evaluation. To this end, we calculated correlation coefficients of the UCP-LFA signals of the MTB infected cohort of *n* = 75 rhesus macaques from Figure 1 to the lung pathology scores (lung PA) or the sum of pulmonary and extrapulmonary disease scores (total PA) (Figure 3). The best correlation between disease and diagnostic signal was observed for IP-10, yielding an R^2^ value of 0.728 (*p* < 0.0001) for total PA and of 0.716 (*p* < 0.0001) for lung PA (Figure 3b). IL-6 and SAA1 showed somewhat weaker yet highly significant correlations with R^2^ = 0.591 (*p* < 0.0001) and 0.582 (*p* < 0.01), respectively, for total PA, and R^2^ = 0.486 (*p* < 0.0001) and 0.449 (*p* < 0.0001), respectively, for lung PA, (Figure 3a,c). When calculating the combined 3BM signature at endpoint, it correlated well with pathology (R^2^ = 0.710 (*p* < 0.0001) for total PA; R^2^ = 0.627 (*p* < 0.0001) for lung PA) (Figure 3d).

### 3.5. Verification of the Diagnostic Potential of UCP-LFAs under Experimental Conditions

As introduced before, our biobank of sera contained specimens from TB research studies with different objectives relating to NHP modeling and disease susceptibility or experimental vaccination. In one of those studies, we explored two cohorts of Indian- versus Chinese-type rhesus macaques that differ in TB disease susceptibility and that were either or not vaccinated with a standard human dose of BCG via the intradermal route, while one treatment group of Indian-type rhesus received BCG via the pulmonary mucosa (Appendix A; [44]). Thus, we evaluated the performance of SAA1, IP-10, and IL-6 UCP-LFA over time in 5 experimental groups of Indian- or Chinese-type rhesus macaques either or not vaccinated with intradermal or mucosal BCG. We have plotted Kaplan–Meier curves of the percentage positivity per group in Figure 4. Interestingly, the most rapid diagnostic kinetics are observed for SAA1, IP-10, and IL-6 in the unvaccinated Indian and Chinese type rhesus macaques (Figure 4). While the Kaplan–Meier curves (representing the binary outcome of diagnostic testing, positive or negative) appear indistinguishable, the absolute UCP-LFA signals over time corroborate the higher TB disease susceptibility of Indian- over Chinese-type rhesus macaques (Appendix A). The results confirm the earlier notion that SAA1 appears as a very early surrogate marker of disease, yielding significantly different levels between the two rhesus genotypes at week three postinfection already, while IP-10 and IL-6 show discriminatory potential at later time points (Appendix A). Indian-type rhesus vaccinated with standard intradermal BCG, which had been shown previously to lack protective signals of reduced pathology scores, yielded UCP-LFA diagnostic profiles with comparably rapid kinetics as the unvaccinated groups. In contrast, the superior protective efficacy of mucosal BCG in this cohort is reflected by the slowest Kaplan–Meier kinetics for all three biomarkers indeed (Figure 4). For BCG vaccinated Chinese type rhesus macaques (intermediately), delayed diagnostic positivity by SAA1, IP-10, and IL-6 signals are in line with the partial efficacy of standard BCG vaccination in this cohort of animals (Figure 4; [44]). The 3BM aggregate score over time for each of the five treatment groups is captured in a heatmap shown in Appendix A, illustrating the potential of these UCP-LFA modalities to distinguish natural or vaccine-modulated disease susceptibility.

In another NHP TB vaccine study, we had demonstrated long-term protective effects of prior BCG up to one year after single high-dose infection (F.V. et al., (BPRC, Rijswijk, The Netherland) unpublished data; Appendix A), by the significant delay in reaching a humane endpoint (time-to-humane-endpoint, or survival) between vaccinees versus unvaccinated controls as well as by the significant reduction of lung and total pathology scores (Appendix A). Here, we analyzed longitudinal serum samples at early time points up to 12 weeks postinfection by UCP-LFA. Again, already from 3 weeks postinfection onwards, we find statistically significant increases of SAA1, IP-10, and IL-6 LFA ratios (although now there is no evidence of SAA1 performing better than the others as an early diagnostic marker), while prior BCG vaccination significantly suppresses these diagnostic signals (Appendix A). When disregarding vaccine treatment, plotting SAA1, IP-10, and IL-6 ratios, and illustrating 3BM in a heatmap by those that reach a premature humane endpoint (HE) versus those that survive up to a year postinfection (endpoint by experimental study design, EE), significant diagnostic positivity at early time points predicts the outcome of experimental infection at the longer term (Figure 5 and Appendix A). Animals protected by prior BCG typically fail to pass the diagnostic threshold (except a few IL-6 responders; Figure 5). To further illustrate the diagnostic potential of our UCP-LFA modalities, we classified animals according to the absolute LFA ratios per time point into tertiles from high, over medium, to low responders and, subsequently, plotted the ensuing Kaplan–Meier curves of the percentage of survival (read: not reaching a humane endpoint). Without exception, the high and medium responders showed a worse prognosis by this time-to-endpoint analysis compared to the low-responder group (Figure 6). Again, the data confirm the diagnostic potential of SAA1, IP-10, and IL-6 UCP-LFA.

Finally, we assessed the performance of these UCP-LFAs in monitoring anti-TB drug treatment in NHP by exploiting banked serum samples from (Indian-type) rhesus macaques that had been infected with a single dose of 15 CFU of MTB, treated four weeks later with rifampin and isoniazid (RIF + INH) for another eight weeks, until study endpoint at twelve weeks after infection (Appendix A). 

As shown in Figure 7, levels of the three diagnostic markers were overall higher in untreated rhesus macaques and rapidly decreased following antibiotic treatment from four weeks postinfection onward. This indicates that UCP-LFA can sensitively detect the host response upon anti-TB treatment and can potentially serve as a tool for early treatment responsiveness.

## 4. Discussion

Minimally invasive tests that allow rapid monitoring of quantitative levels of biomarkers reflecting TB-associated pathology could facilitate screening for TB during maintenance of valuable NHP colonies and experimental NHP models for the development of TB vaccines and anti-TB therapy. In this exploratory study, we validated previously developed user-friendly lateral flow assays (UCP-LFAs), designed to quantitatively detect human proteins associated with active TB [26,31,33,34,50], for use in NHPs. Our data show that levels of SAA1, IL-6, and IP-10 detected by UCP-LFAs correlate with MTB infection load and disease severity as determined by pathology scoring. Moreover, these three biomarkers combined (3BM) could sensitively detect a reduction of TB disease in macaques as a result of prophylactic BCG vaccination or drug treatment, underpinning the quantitative character of these host response markers for diagnostic purposes. 

SAA1 has been identified as a potential biomarker for active TB both in humans and NHPs [51,52,53]. IP-10 is a mechanistic marker in trained immunity to MTB [54], which has shown diagnostic value for infection with MTB in humans [55], and was induced significantly by PPD stimulation in the whole blood of NHPs with TB [56]. Furthermore, IL-6 has shown potential as a biomarker for a triage test for active pulmonary TB in adults with persistent cough [57,58] and as a biomarker for monitoring treatment efficacy of active TB disease [59]. 

In our study, both IL-6 and SAA1 levels in sera of uninfected rhesus macaques were undetectable, whereas MTB infection resulted in a dose-dependent increase of these proteins, making these proteins sensitive markers of infection which can be detected by UCP-LFAs. Levels of IP-10, on the other hand, still rapidly increasing upon infection, were already detectable in uninfected, healthy animals as well. This indicates that in nonexperimental settings like regular NHP colony screening, a cut-off threshold for IP-10 is required to allow detection of MTB infection. Of note is that overall the levels of IP-10 and SAA1 increased faster than IL-6; a significant increase in IP-10 and SAA1 was already observed in all study cohorts at week three postinfection. Moreover, IP-10 levels correlated stronger with both disseminated and lung PA scores, whereas SAA1 and IL-6 levels appeared better at predicting TB lung pathology. Since the correlation between the 3BM was not strong, added diagnostic value could be obtained by combined assessment of all three markers, especially in the initial weeks after infection. The use of the 3BM signature improved diagnostics value compared to single UCP-LFAs only (ROC-AUC and sensitivity). Regarding the observed sensitivity in this study, it is of note that we used a selected set of samples, which implies a limitation of the interpretation of our results. Follow-up screening in a larger NHP colony can further verify the specificity of this UCP-LFA-based method.

With respect to the 3BM signature, we recently developed a multibiomarker test (MBT) which measures six biomarkers simultaneously in one sample, thus accommodating the detection of biomarker signatures that synergistically have a higher sensitivity and specificity than the single markers [28]. The MBT thereby provides the basis for immunodiagnostic POC tests for various (infectious) diseases and use cases by integrating biomarker profiles into multiplexed diagnostic tools. Moreover, the flexible MBT [28] format allows the exchange of biomarkers in case additional more potent markers for TB in NHP are identified.

We have demonstrated a good association between protein levels detected by UCP-LFA and MTB-induced pathology in experimentally infected NHPs. LFA performance in NHP upon ultralow-dose infection and latent TB [60], or upon natural infection, however, remains to be established. In that respect, since TB outbreak in an unvaccinated colony may show rapid disease progression, combined screening using both IGRAs for latent infection as well as UCP-LFA for host proteins IP-10, SAA1 and IL-6 could be (further developed into) a valuable diagnostic platform for rapid screening of NHP. Meanwhile, more research will be needed to establish the diagnostics applicability of UCP-LFA in experimental NHP TB scenarios. With regard to such experimental settings, we anticipate that, typically, more than one assay shall be exploited to monitor the outcome of infection and treatment rather than using UCP-LFA on its own (either in its current form or in any future configuration of multiple analytes).

Nevertheless, UCP-LFA were able to detect relatively minor differences in pathology induced in rhesus macaque genotypes with different susceptibility for TB after infection with the same dose. In conjunction with the more severe TB disease phenotype observed in the Indian-type rhesus macaques [12,44], biomarker levels increased more rapidly in this group compared to the Chinese-type rhesus macaques. Furthermore, the effect of BCG vaccination and treatment in this cohort could be monitored effectively by UCP-LFA, as the levels of the markers correlated very well with the outcome of infection (pathology, time-to-humane endpoint) in these animals. The current trend in the vaccine research and development field in NHP is moving towards (ultra) low-dose or repeated limiting dose challenge and low burden (minimal) disease readout. Thus, one can anticipate that studies addressing (therapeutic suppression of) reactivation of disease over more extended periods might benefit from supportive serological LFA diagnostics at regular intervals.

In humans, TB C-reactive protein (CRP) represents a biomarker with potential for detecting active TB, even in HIV-positive individuals [61]. Likewise, we have established the value of CRP as a marker of TB disease (severity) in NHP (vaccine) studies before [62]. In our first attempt with UCP-LFA, we could detect CRP in a few rhesus macaques with severe TB, but the performance was insufficient to warrant further analysis in this study. Higher homology of nucleotide and protein in NHPs versus humans does increase the chance of successful application in NHP of tests based on human tools. For example, we could not detect SAA1, IL-6, IP-10, and CRP in marmoset (data not shown) like in rhesus macaques. Compared to the marmoset (percent identity of nucleotide and protein were 84.43–89.86% and 79.38–89.15%, respectively), the rhesus macaques have higher homology (91.38–95.00%, 87.70% > 96.70%, respectively) to humans. We were unable to predict whether UCP-LFA would work based on homology. In fact, the 11 proteins we tried to detect by UCP-LFA in rhesus macaques all have high homology of nucleotide (>91.01%) and protein (>86.96%) in humans and rhesus macaques (Appendix A). This indicates that mAbs used in this UCP-LFA differ in their ability to bind to cross-species conserved epitopes, thereby demonstrating the importance of mAb selection in UCP-LFA development for NHP. Further exploration of other mAbs is required to expand the signature for sensitive detection of TB in NHP and subsequent application to our MBT platform [28].

UCP-LFAs are compatible with the use of fingerprick blood in both humans [50] and even across a species barrier in *M. leprae*-infected red squirrels [63]. Additionally, in research settings, it may support the readout of experimental infection. The user-friendly and low resource-requiring methodology of UCP-LFAs using fingerprick blood may thus provide a rapid tool for colony screening and maintenance purposes, especially in resource and infrastructure limited settings.

## 5. Conclusions

This exploratory study shows that the UCP-LFAs developed for use with human samples can be applied to detect SAA1, IL-6, and IP-10 in serum of macaques. Significantly, the levels of these biomarkers after experimental MTB infection correlate with disease outcomes in these animals. Thus, these results establish the potential of UCP-LFAs as a low-cost, rapid and user-friendly tool that can be applied in experimental macaque studies on TB vaccine research and drug development as well as in detecting (nonexperimental) MTB infection in colonies.

## Figures and Tables

**Figure 1 biology-10-01260-f001:**
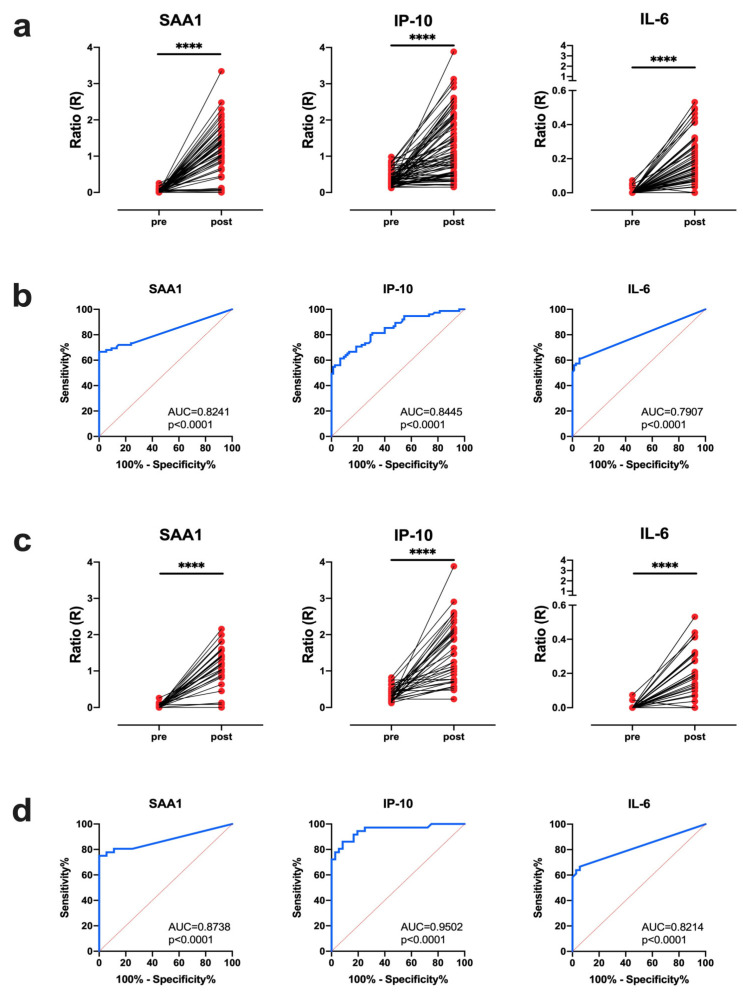
Distinctive increase in serum SAA1, IP-10 and IL-6 after MTB infection. SAA1, IP-10 and IL-6 levels were measured by UCP-LFA in serum from MTB Erdman (500 CFU, *n* = 53; 15 CFU, *n* = 12; 1-7 CFU, *n* = 10) infected rhesus macaques with or without BCG pre-vaccination (*n* = 75, Appendix A). Endpoints varied between 6 weeks and 52 weeks after infection. Results are displayed as the Ratio value (R) between Test (T) and Flow-Control (FC) signal based on relative fluorescence units (RFUs; excitation at 980nm and emission at 550 nm) measured at the respective lines. (**a**,**b**) Analysis of three biomarker levels pre and postinfection in MTB Erdman infected rhesus macaques (*n* = 75). (**c**,**d**) Analysis in untreated, infected control animals from high- (500 CFU, *n* = 24) and low-dose (15 CFU, *n* = 12) MTB infection studies (*n* = 36). (**a**,**c**) Significant differences between biomarker levels (*y*-axis) pre and postinfection were determined by Wilcoxon matched-pairs signed rank tests. (**b**,**d**) The ability to distinguish pre from postinfection states in the same animals was evaluated by ROC curve analysis, including Area Under the Curve (AUC) measurements (ROC-AUC). AUC: area under the curve; *p*-values: **** *p* < 0.0001.

**Figure 2 biology-10-01260-f002:**
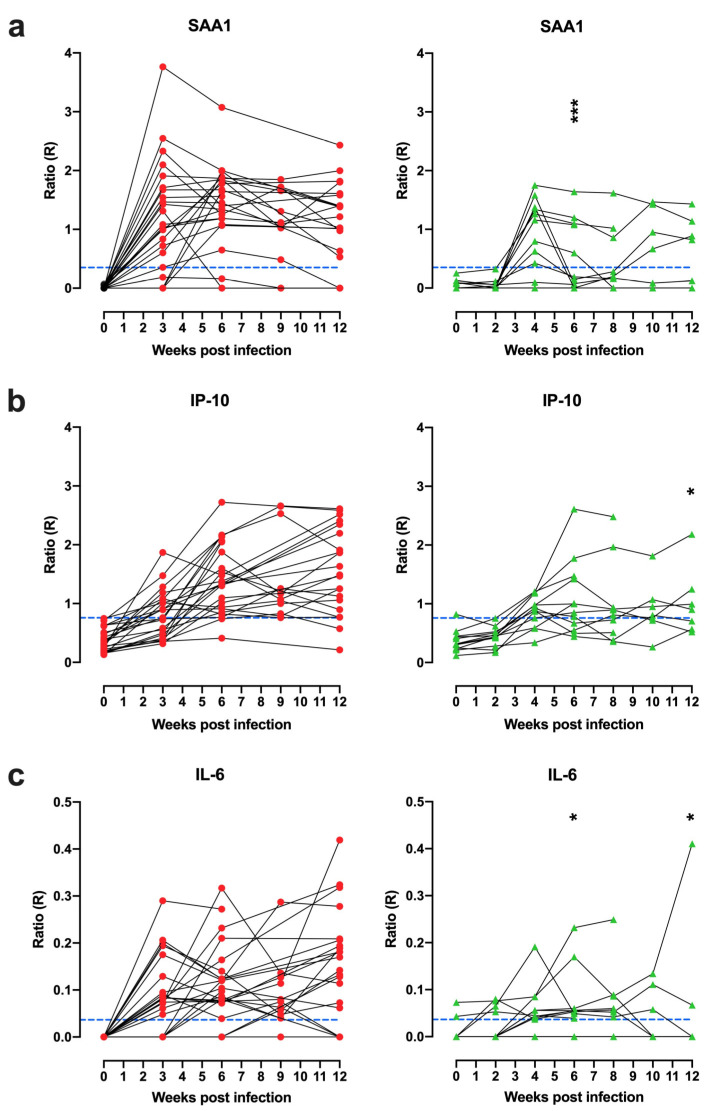
SAA1, IP-10, and IL-6 serum levels increased after MTB infection. SAA1, IP-10 and IL-6 levels were measured by UCP-LFA in serum from high-dose (500 CFU, *n* = 24, red dots; left panels) or low-dose (15 CFU, *n* = 12, green triangles; right panels) MTB Erdman infected rhesus macaques (*n* = 36) at different timepoints (*x*-axis) pre and postinfection. Cutoff values for positivity (blue dashed lines) were calculated by applying the Youden’s index as described in Table 1b. (**a**–**c**) The levels of SAA1, IP-10, and IL-6 increase after MTB infection. Differences between high and low doses at 3/4, 6, and 12 weeks postinfection were determined by Mann–Whitney U tests indicated in the right panel. *p*-values: * *p* < 0.05, *** *p* < 0.001.

**Figure 3 biology-10-01260-f003:**
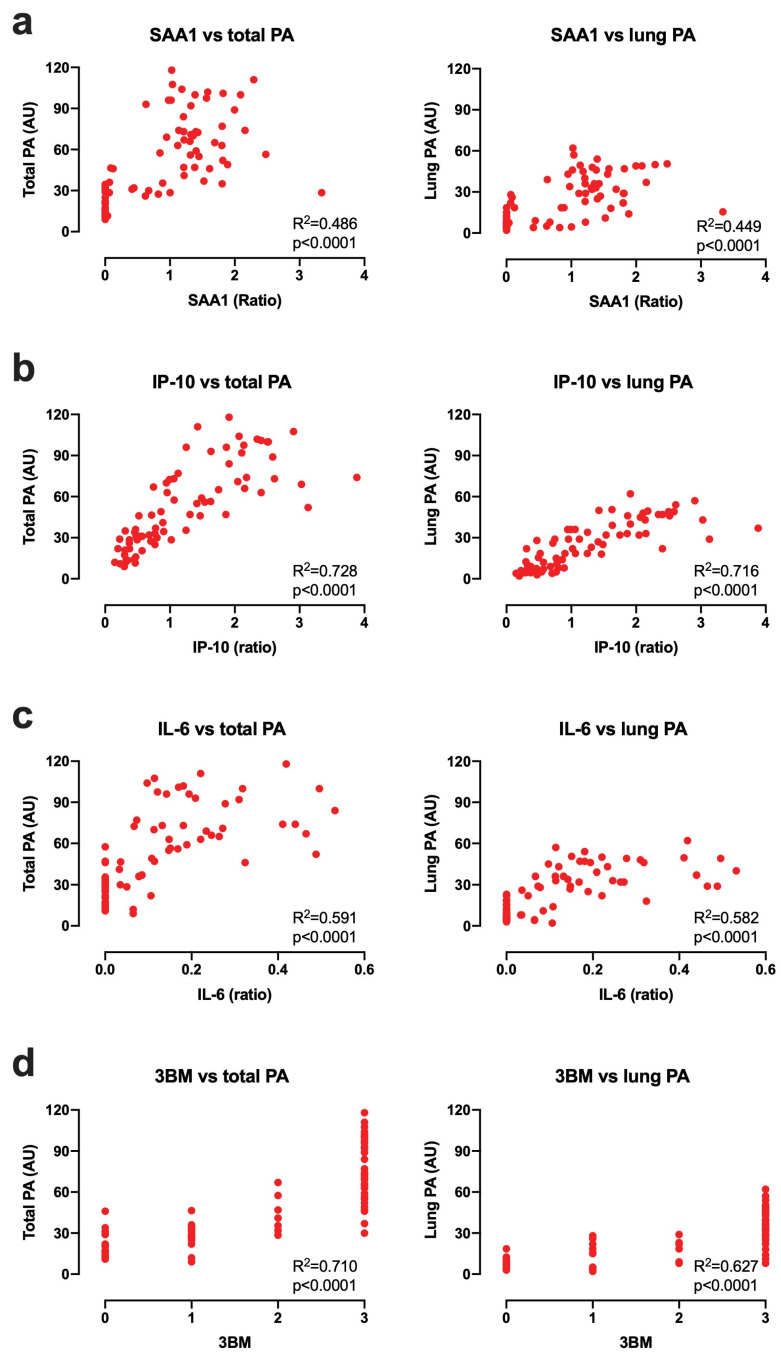
Correlation between SAA1, IP-10, and IL-6 levels and total/ lung pathology score at the endpoint. (**a**) SAA1, (**b**) IP-10, and (**c**) IL-6 levels (*x*-axis) were measured by UCP-LFA in serum from MTB Erdman infected rhesus macaques (*n* = 75, see Figure 1). Tuberculosis pathology (PA; *y*-axis) scoring methods are described in Materials and Methods (2.2), and PA is expressed as arbitrary units (AU). R^2^ is the square of the Spearman correlation coefficient. (**d**) Values above the cutoff per biomarker (described in Table 1b) were considered positive. 3BM (*x*-axis) was generated using the sum of all positive tests results from the individual markers. (**a**–**d**) SAA1, IP-10, IL-6, and 3BM showed a good correlation with total- and lung pathology scores at the endpoint.

**Figure 4 biology-10-01260-f004:**
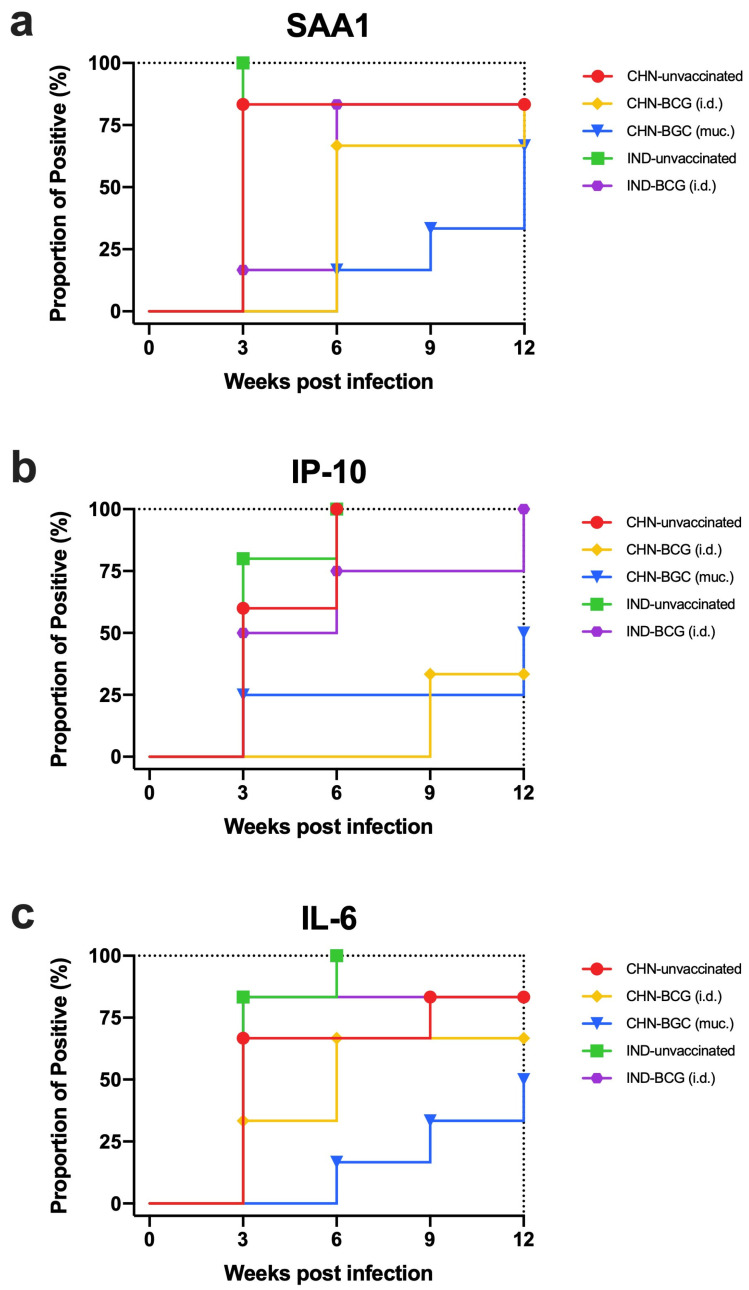
SAA1, IP-10, and IL-6 levels in two rhesus genotypes with/without BCG vaccination after MTB infection. SAA1, IP-10, and IL-6 levels were measured by UCP-LFA in serum from high-dose (500 CFU) MTB Erdman infected Chinese-type (CHN, *n* = 18) and Indian-type (IND, *n* = 12) rhesus macaques. 12 macaques were unvaccinated (CHN, *n* = 6, red dots; IND, *n* = 6, green squares); 18 were vaccinated with BCG 17 weeks before infectious challenge with MTB, of which 12 by intradermal injection (BCG i.d.; CHN, *n* = 6, yellow rhombus, and IND, *n* = 6, purple hexagons) and 6 by pulmonary mucosal administration (BCG muc; CHN, *n* = 6, blue triangles)) (**a**–**c**) Kaplan–Meier curves were plotted, showing the proportion of positive NHP over time for the Chinese vs. Indian rhesus macaques with/without BCG vaccination.

**Figure 5 biology-10-01260-f005:**
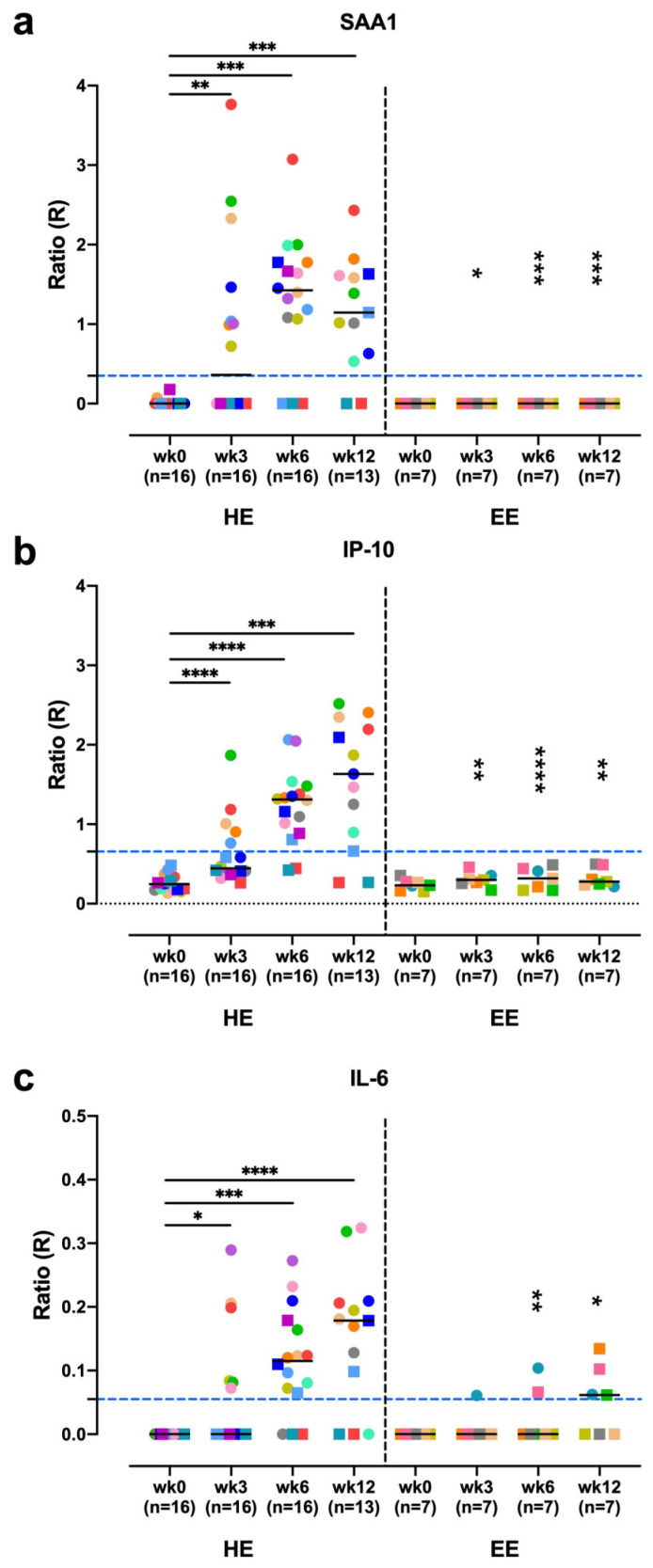
SAA1, IP-10, and IL-6 levels in serum increased after infection and are associated with the disease severity. SAA1, IP-10, and IL-6 levels (*y*-axis) were measured by UCP-LFA in serum from high-dose (500 CFU) MTB Erdman infected rhesus macaques (*n* = 23), of which 16 animals developed progressive disease and reached a premature humane endpoint (HE), while the endpoint by protocol was fixed at 50 weeks after challenge (EE). The median values of each group are indicated by horizontal lines. Cut-off for positivity (blue dashed lines) as described in Table 1b. Each coloured square (BCG-vaccinated, *n* = 12) or dot (unvaccinated, *n* = 11) represents an individual NHP. (**a**–**c**) SAA1, IP-10, and IL-6 postinfection levels were significantly increased in the humane endpoint (HE) group only. SAA1, IP-10 and IL-6 levels in NHPs with the HE in week 3, 6 (*n* = 16) and week 12 (*n* = 13) were higher than the EE (*n* = 7). Significant differences between each timepoint versus week 0 were determined by Wilcoxon matched-pairs signed rank tests. Since three macaques reached a HE before week 12 in the HE group, only 13 data points are available for week 12. Differences between HE versus EE at per time point were determined by Mann-Whitney U tests indicated in the right panel (EE). *p*-values: * *p* < 0.05, ** *p* < 0.01, *** *p* < 0.001, **** *p* < 0.0001.

**Figure 6 biology-10-01260-f006:**
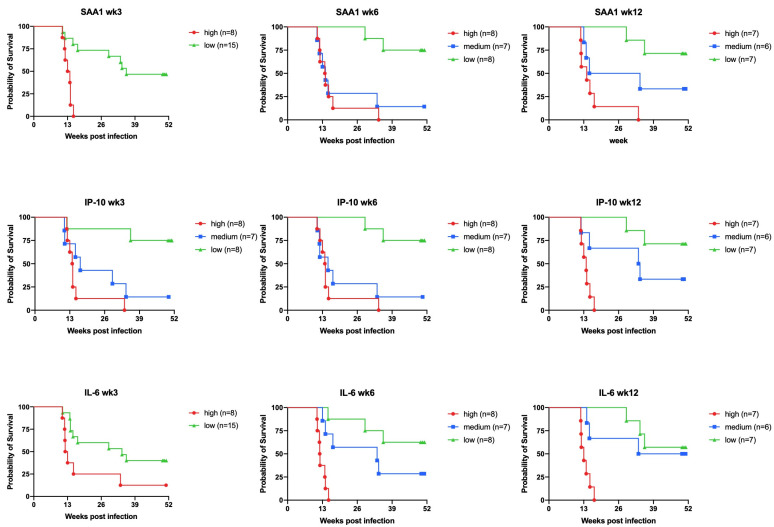
Correlation between SAA1, IP-10, and IL-6 levels in serum and disease severity. SAA1, IP-10, and IL-6 levels were measured by UCP-LFA in serum from high-dose (500 CFU) MTB Erdman infected rhesus macaques (*n* = 23). (Since three rhesus macaques reached a humane endpoint before week 12, only 20 instead of 23 measurements are depicted at week12). The animals were sorted according to the protein levels at each time point. The top 33% of the value is defined as high responders, the middle 33% is defined as medium responders, and the last 33% is defined as low responders. All animals with undetectable protein levels are defined as low responders. Kaplan–Meier curves were plotted of individual endpoints for the high vs. the medium vs. the low responders after MTB infection. High responders after MTB infection arrive at HE earlier than medium or low responders.

**Figure 7 biology-10-01260-f007:**
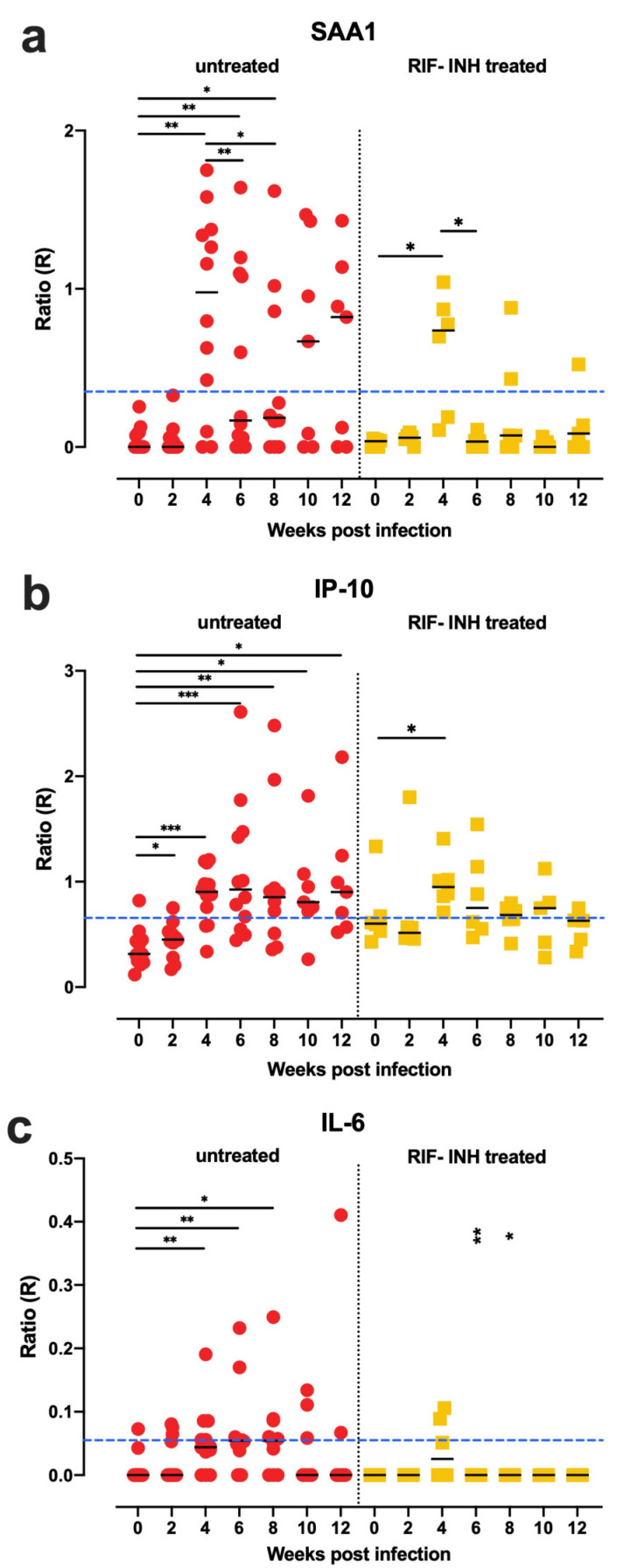
SAA1, IP-10, and IL-6 decreased to the baseline levels upon antibiotic treatment. SAA1, IP-10, and IL-6 levels (*y*-axis) were measured by UCP-LFA in serum from low-dose (15 CFU) MTB Erdman infected rhesus macaques (*n* = 18), six of them were treated by rifampin (RIF) and isoniazid (INH) from week four postinfection for eight weeks until endpoint at week 12 twelve The median values of each group are indicated by horizontal lines. Red dots indicate untreated macaques; yellow squares indicate RIF + INH treated macaques. Cutoff (blue dashed lines) applied was described in Table 1b. (**a**–**c**) The SAA1, IP-10, and IL-6 levels were decreased after RIF + INH treatment. Significant differences between each timepoint versus week 0 were determined by Wilcoxon matched-pairs signed rank tests; differences between untreated versus RIF + INH treated at per time point were determined by Mann-Whitney U tests indicated in the right panel (RIF + INH). *p*-values: * *p* < 0.05, ** *p* < 0.01, *** *p* < 0.001.

**Table 1 biology-10-01260-t001:** Assay performance characteristics of UCP-LFAs in MTB infected rhesus macaques.

(**a**). Assay performance to discriminate pre and postinfection (*n* = 75)							
	**Cut off**	**Sensitivity (%)**	**95% CI (%)**	**Specificity (%)**	**95% CI (%)**	**AUC-ROC**	***p* Value**
SAA1	R > 0.336	66.67	55.42 to 76.29	100	95.13 to 100.0	0.8241	<0.0001
IP-10	R > 0.766	61.33	50.02 to 71.54	93.33	85.32 to 97.12	0.8445	<0.0001
IL-6	R > 0.034	60	48.69 to 70.34	94.67	87.07 to 97.91	0.7907	<0.0001
3BM	#pos BM ≥ 1	78.67	68.12 to 86.42	89.33	80.34 to 94.50	0.8694	<0.0001
(**b**). Assay performance to discriminate pre and postinfection (*n* = 36)							
	**Cut off**	**Sensitivity (%)**	**95% CI (%)**	**Specificity (%)**	**95% CI (%)**	**AUC-ROC**	***p* Value**
SAA1	R > 0.350	75.00	58.93 to 86.25	100	90.36 to 100.00	0.8738	<0.0001
IP-10	R > 0.656	86.11	71.34 to 93.92	91.67	78.17 to 97.13	0.9502	<0.0001
IL-6	R > 0.055	63.89	47.58 to 77.52	97.22	85.83 to 99.86	0.8214	<0.0001
3BM	#pos BM ≥ 1	86.11	71.34 to 93.92	88.89	74.69 to 95.59	0.9150	<0.0001

SAA1, IP-10 and IL-6 levels were measured by UCP-LFAs in serum from MTB Erdman infected rhesus macaques (*n* = 75) before infection and at the study endpoint. Results are displayed as the Ratio value (R) between Test (T) and Flow-Control (FC) signal based on relative fluorescence units (RFUs; excitation at 980 nm and emission at 550 nm) measured at the respective lines on the UCP-LF strips. The ability to distinguish pre from postinfection states in the same animals was evaluated by ROC curve analysis, and the corresponding Area Under the Curve (AUC) was determined (ROC-AUC). The cut-off value for each biomarker was determined by Youden’s index; values above the cutoff for each biomarker were considered positive. A 3BM (3 biomarkers) signature was generated using the sum of all positive tests results from the individual markers. (**a**) Assay performance in MTB Erdman infected rhesus macaques (*n* = 75). (**b**) Assay performance in unvaccinated rhesus macaques only (*n* = 36), infected with a high- (500 CFU, *n* = 24) or a low-dose (15 CFU, *n* = 12) of MTB Erdman. CI, confidence interval.

**Table 2 biology-10-01260-t002:** Different assay performance of UCP-LFAs in high-dose and low-dose MTB infected rhesus macaques.

(**a**). AUCs of high-dose MTB infected RMs (*n* = 24)						
	**wk3**		**wk6**	**wk9**		**wk12**
SAA1	0.903		0.951	0.894		0.889
IP-10	0.858		0.984	1.000		0.956
IL-6	0.854		0.938	0.955		0.905
3BM	0.948		0.997	0.996		0.968
(**b**). AUCs of low-dose MTB infected RMs (*n* = 12)						
	**wk2**	**wk4**	**wk6**	**wk8**	**wk10**	**wk12**
SAA1	0.531	0.868	0.747	0.763	0.750	0.774
IP-10	0.646	0.944	0.944	0.883	0.870	0.952
IL-6	0.597	0.795	0.799	0.783	0.655	0.571
3BM	0.583	0.8958	0.875	0.858	0.964	0.821

SAA1, IP-10 and IL-6 levels were measured by UCP-LFA before and at several time points after (**a**) high-dose (500 CFU, *n* = 24) or (**b**) low-dose (15 CFU, *n* = 12) MTB Erdman infection. The ability to distinguish pre from postinfection states was evaluated per marker and for the 3BM (3 biomarkers) signature by ROC curve analysis over time, and the corresponding Area Under the Curve (AUC) was determined (ROC-AUC). Values above the cutoff per biomarker (described in Table 1b) were considered positive. 3BM signature was generated using the sum of all positive tests results from the individual markers. wk, week; RM, rhesus macaque; BM, biomarker.

## Data Availability

Not applicable.

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
