# Peer review of "Quantitative Rapid Test for Detection and Monitoring of Active Pulmonary Tuberculosis in Nonhuman Primates"

_biology, 2021, doi:10.3390/biology10121260_

Round 1

Reviewer 1 Report

The authors have made corrections to be responsive to reviewers, while positioning the work more properly as exploratory and identifying next steps for assay development.

Author Response

Reviewer 1:

Comments and Suggestions for Authors

The authors have made corrections to be responsive to reviewers, while positioning the work more properly as exploratory and identifying next steps for assay development.

Reviewer 2:

Comments and Suggestions for Authors

Materials and Methods

Line 133 The 2.2. NHP Study Cohorts states that low-dose infections is a low-dose of 1 to 7 CFU.

This is different for the low infection dose in experimental results. Such as line of 216 “low-dose (15 CFU, n=12)” why 1-7 and 15 CFU are chosen as the low infection dose in the article, Give an explanation.

Reply:

As indicated correctly by the reviewer, we have applied different challenge doses.

Therefore, we have now defined 15 CFU as “low-dose” and 1-7 CFU as “ultra-low-dose”. Has been changed in the Materials and Methods (Line 133) and Table S5 in supplementary file.

Result

The authors had presented their data without making such vague description, this would be a study appropriate for publication.

Figure 1

1.Figure legend for Fig 1 is confusing.

e.g “(a, b) SAA1, IP-10 and IL-6 levels were measured by UCP-LFA in serum from Mtb Erdman infected rhesus macaques (n=75). Endpoints varied between 6 weeks and 52 weeks after infection. Results are displayed as the Ratio value (R) between Test (T) and Flow-Control (FC) signal based on relative fluorescence units (RFUs; excitation at 980nm and emission at 550 nm) measured at the respective lines (y-axis).”

Obviously, this is a description about a. How about b?

“(c, d) The same analysis in non-treated, infection control animals from high- (500 CFU, n=24) and low-dose (15 CFU, n=12) Mtb infection studies.”

Figure 1(c d) did not use the same statistical method, this explanation is confusing. Which part represents the result of a high infection dose, and which part is the result of a low infection dose?

The authors need to correct this inaccuracy demonstrate.

We corrected the legend of Figure 1 as follows:

Figure 1. Distinctive increase in serum SAA1, IP-10 and IL-6 after Mtb infection.

SAA1, IP-10 and IL-6 levels were measured by UCP-LFA in serum from Mtb Erdman (500 CFU, n=53; 15 CFU, n=12; 1-7 CFU, n=10) infected rhesus macaques with or without BCG pre-vaccination (n=75, Table S1-5). Endpoints varied between 6 weeks and 52 weeks after infection. Results are displayed as the Ratio value (R) between Test (T) and Flow-Control (FC) signal based on relative fluorescence units (RFUs; excitation at 980nm and emission at 550 nm) measured at the respective lines. (a, b) Analysis of three biomarker levels pre- and post-infection in Mtb Erdman infected rhesus macaques (n=75). (c, d) Analysis in non-treated, infected control animals from high- (500 CFU, n=24) and low-dose (15 CFU, n=12) Mtb infection studies (n=36). (a, c) Significant differences between biomarker levels (y-axis) pre- and post-infection were determined by Wilcoxon matched-pairs signed rank tests. (b, d) The ability to distinguish pre- from post-infection states in the same animals was evaluated by ROC curve analysis, including Area Under the Curve (AUC) measurements (ROC-AUC). AUC: area under the curve; P-values: *p<0.05, **p<0.01, ***p<0.001, ****p<0.0001.

Line 249 “3.2. Diagnostic Performance Relative to Infection Dose.”

The experimental design lacks a blank control group.

Reply: The aim of this section is to compare the assay performances to discriminate pre- and post-infection status using different doses of infection. The assay performance data from blank (uninfected) controls, alas, are not available. We do see the reviewer's point, but must indicate that from an ethical perspective it would be very difficult to justify sampling of non-infected controls at harmonised time points in the exploratory settings of our first experiments on LFA and NHP TB samples.

line 213.“ Endpoints varied between 6 weeks and 52 weeks after infection.”

In the text, there is clear finding that these biomarkers showed highest levels in high-dose infected rhesus macaques with significant difference time. Therefore, it is unreasonable for this experiment to use the values at different endpoints from 6 to 52 weeks as the post-infection statistics.

Reply: Since our samples come from the biobank of different-setting experiments, there is not a (common) time point that all animals are sampled. We can only choose the last time point as post-infection. Moreover, we used a Wilcoxon matched-pairs signed rank test to analyze the performance of distinguishing pre- and post-infection status, such that each animal is only compared with itself before infection, instead of cross-compared with other animals. Therefore, the impact of different infection intervals is limited. Different animals have different end points, which does not affect the comparison of pre- and post- infection

Line 249 “3.2. Diagnostic Performance Relative to Infection Dose

Reply: see above.

Figure legend for Fig 3 is not descriptive enough.

Reply: We have adapted the legend as follows and indicated that further explanation can be found in the methods.

Figure 3. Correlation between SAA1, IP-10 and IL-6 levels and total/ lung pathology score at the endpoint.

(a) SAA1, (b) IP-10 and (c) IL-6 levels (x-axis) were measured by UCP-LFA in serum from Mtb Erdman infected rhesus macaques (n=75, see Figure 1). Tuberculosis pathology (PA; y-axis) scoring methods are described in Materials and Methods (2.2) and PA is expressed as arbitrary units (AU). R2 is the square of the Spearman correlation coefficient. (d) Values above the cut-off per biomarker (described in Table 1b) were considered positive. 3BM (x-axis) was generated using the sum of all positive tests results from the individual markers. (a-d) SAA1, IP-10, IL-6, and 3BM showed a good correlation with total- and lung pathology scores at the endpoint.

Line 273. “Of note, the three biomarkers showed significant but not particularly high correlations with each other (Figure S4).”

Line 453. “Additionally, we observed that the levels of SAA1, IP-10 and IL-6 were not correlated strongly, ”

It is recommended to delete 453 lines of content.

Reply: We deleted the line 453 and adapted the sentence as follows (line 456):

Since correlation between the 3BM was not strong, added diagnostic value could be obtained by combined assessment of all three markers, especially in the initial weeks after infection the use of the 3BM signature, improved diagnostics value compared to single UCP-LFAs only (ROC-AUC and sensitivity).

Reviewer 2 Report

Materials and Methods

Line 133  The 2.2. NHP Study Cohorts states that low-dose infections is a low-dose of 1 to 7 CFU .

This is different for the low infection dose in experimental results. Such as line of 216 “low-dose (15 CFU, n=12)” why 1-7 and 15 CFU are chosen as the low infection dose in the article, Give an explanation.

Result

 The authors had presented their data without making such vague description, this would be a study appropriate for publication.

Figure 1

1.Figure legend for Fig 1 is confusing.

e.g “(a, b) SAA1, IP-10 and IL-6 levels were measured by UCP-LFA in serum from Mtb Erdman infected rhesus macaques (n=75). Endpoints varied between 6 weeks and 52 weeks after infection. Results are displayed as the Ratio value (R) between Test (T) and Flow-Control (FC) signal based on relative fluorescence units (RFUs; excitation at 980nm and emission at 550 nm) measured at the respective lines (y-axis).”

Obviously, this is a description about a. How about b?

“(c, d) The same analysis in non-treated, infection control animals from high- (500 CFU, n=24) and low-dose (15 CFU, n=12) Mtb infection studies.”

Figure 1( c d ) did not use the same statistical method, this explanation is confusing. Which part represents the result of a high infection dose, and which part is the result of a low infection dose? The authors need to correct this inaccuracy demonstrate.

  1. Line 249 “3.2. Diagnostic Performance Relative to Infection Dose.”

The experimental design lacks a blank control group.

  1. line 213.“ Endpoints varied between 6 weeks and 52 weeks after infection.”

In the text, there is clear finding that these biomarkers showed highest levels in high-dose infected rhesus macaques with significant difference time. Therefore, it is unreasonable for this experiment to use the values at different endpoints from 6 to 52 weeks as the post-infection statistics.

Line 249 “3.2. Diagnostic Performance Relative to Infection Dose

  1. Figure legend for Fig 3 is not descriptive enough.

  2. Line 273. “Of note, the three biomarkers showed significant but not particularly high correlations with each other (Figure S4).”

Line 453. “Additionally, we observed that the levels of SAA1, IP-10 and IL-6 were not correlated strongly, ”

It is recommended to delete 453 lines of content.

Author Response

(The authors gave the same response as above.)
